Subject Areas:
ecology/cognition/behaviour

Keywords:
movement ecology, cognition, spatial memory, accuracy, speed, straightness

Author for correspondence:
Christine E. Beardsworth
e-mail: c.e.beardsworth@gmail.com

# Spatial cognitive ability is associated with transitory movement speed but not straightness during the early stages of exploration

Christine E. Beardsworth[1], Mark A. Whiteside[1,2], Lucy A. Capstick[1], Philippa R. Laker[1], Ellis J. G. Langley[1], Ran Nathan[3], Yotam Orchan[3], Sivan Toledo[4], Jayden O. van Horik[1] and Joah R. Madden[1]

[1]Centre for Research in Animal Behaviour, Psychology, University of Exeter, Exeter EX4 4QG, UK
[2]School of Biological and Marine Sciences, University of Plymouth, Drake Circus, Plymouth PL4 8AA, UK
[3]Movement Ecology Laboratory, The Alexander Silberman Institute of Life Sciences, The Hebrew University of Jerusalem, Jerusalem 91904, Israel
[4]Blavatnik School of Computer Science, Tel-Aviv University, Tel Aviv 67798, Israel

 CEB, 0000-0003-1308-1455; MAW, 0000-0001-7212-7279;
LAC, 0000-0002-0514-5554; PRL, 0000-0002-3214-4098;
EJGL, 0000-0001-8980-8206; RN, 0000-0002-5733-6715;
ST, 0000-0002-9524-7115; JOvH, 0000-0002-8319-911X;
JRM, 0000-0002-0691-0967

Memories about the spatial environment, such as the locations of foraging patches, are expected to affect how individuals move around the landscape. However, individuals differ in the ability to remember spatial locations (spatial cognitive ability) and evidence is growing that these inter-individual differences influence a range of fitness proxies. Yet empirical evaluations directly linking inter-individual variation in spatial cognitive ability and the development and structure of movement paths are lacking. We assessed the performance of young pheasants (*Phasianus colchicus*) on a spatial cognition task before releasing them into a novel, rural landscape and tracking their movements. We quantified changes in the straightness and speed of their transitory paths over one month. Birds with better performances on the task initially made slower transitory paths than poor performers but by the end of the month, there was no difference in speed. In general, birds increased the straightness of their path over time, indicating improved efficiency

independent of speed, but this was not related to performance on the cognitive task. We suggest that initial slow movements may facilitate more detailed information gathering by better performers and indicates a potential link between an individual's spatial cognitive ability and their movement behaviour.

## 1. Introduction

Reducing the time and energy spent searching for resources can help animals maximize their foraging efficiency and reduce their exposure to predators [1], such that even small increases in the efficiency of movements can accumulate across a lifetime and bring fitness benefits [2]. There are marked individual differences in patterns of movement behaviour [3], which can be highly repeatable [4,5]. An individual's spatial cognitive ability, namely, the ability to collect, process, store and use spatial information has been suggested to influence movement decisions [6–8] and improve movement efficiency [2]. This ability can be assayed in both laboratory [9–11] and wild populations [12,13] and performances on tasks assaying spatial abilities have been correlated with proxies of fitness including better survival [14,15], increased sexual success [16,17] or reproductive investment [18,19]. Indirect links have been made between the exploration of a novel environment in the laboratory and spatial cognition at a population level [20]. However, to our knowledge, no studies have yet investigated a link between spatial cognitive ability and the development of movement behaviour of an individual. Specifically, a link between spatial cognitive ability and movement between areas of interest such as foraging patches or refuges may be expected. Previous research shows that the speed and straightness of transitory paths increases with familiarity of the environment [21,22]. Since turning can be costly [23], improving the efficiency of transitory paths may yield energetic benefits and we may expect that differences in spatial cognitive ability could modulate these changes.

Quantifying individual movement in real-world landscapes demands high spatial and temporal resolution tracking. Individuals should be followed continuously over extended time periods and their location logged at time intervals relevant to their typical movement speeds and distances travelled [24,25]. This permits an individual's path to be described both spatially and temporally. If measures from the same individual can be collected repeatedly over time, then their improvements in efficiency, as indicated by a decrease in travelling time, distance or tortuosity of the path between two locations [26], can be established. Crucially, to measure improvement in movement efficiency, it is essential that the prior experience of the individual in that environment is known and accounted for, so that a baseline is established. This means that an individual should be tracked as early as possible from the first time that they enter a landscape, otherwise older or more experienced individuals may appear to be more efficient simply because they have more knowledge of that environment. Additionally, it is desirable to account for the effects that more experienced individuals (e.g. parents) may have on the development of a focal individual's movement. This could be through leadership or followership as naïve individuals may accompany others with prior knowledge of the landscape, which could help them to develop more efficient routes [27,28].

Differences in inter-individual spatial cognitive ability can be assessed under controlled conditions by adapting well-established methods used within comparative psychology. Tasks that simulate foraging can entail food being hidden within a set of potential locations [29,30] frequently arranged as a grid [10] or radial maze structure [31] and performance can be monitored over multiple trials (e.g. [23]). The pheasant, *Phasianus colchicus*, provides an established model for measuring individual differences in cognitive abilities [32], and these differences can have fitness consequences in terms of survival [33]. While evidence from other species suggests that captive and wild populations may differ in other cognitive traits, such as problem solving [34], by using a captive population and rearing birds from hatching, we were able to control for age, experience and environment. Pheasant chicks are precocial and can therefore be reared without parents and under controlled conditions, removing opportunities for inter-generational learning and variation in early life experiences. The annual *en masse* release of pheasants as a managed game bird in the UK provides a rare opportunity to explore how spatial cognitive ability may relate to movement in a free-roaming terrestrial bird. Coupled with their large size and relatively sedentary nature, they can easily be tracked in the wild. Because entire cohorts are released simultaneously and at the same location, all birds are naïve to the environment at the point of release and have equal opportunity to explore the same area with exposure to the same distribution of resources and threats. Pheasants therefore provide a useful model to study the influence of cognitive traits on the development and efficiency of an individual's movement paths in natural landscapes.

We assayed the spatial cognitive ability of a cohort of pheasant chicks in early life using a five-arm radial maze with a single baited arm. Later, we released these birds into the wild with a soft-release protocol [33,35], and monitored their movements using the reverse-GPS system, ATLAS [36,37]. At our study site, these birds are limited in access to informed individuals who they could follow, since adults are caught for breeding and released elsewhere. As individuals become familiar with a landscape, their trajectories between places of interest are expected to increase both in straightness and speed [21,22]. However, we may expect differences in movement traits between individuals of differing spatial cognitive abilities, due to a different strategy [8] or capacity for collecting spatial information. First, if individuals initially explore the environment in similar ways, we may expect birds that perform well in the spatial task to take less time to learn about their environment, resulting in quicker improvements in the speed and straightness of transitory paths. However, more accurate decision making relies on collecting high-quality information which is frequently achieved by increased sampling time, [38–40] and/or making more tortuous movements when gathering information [41]. We may therefore expect birds that perform well in the spatial task to make slower and/or more tortuous paths during the initial stages of exploration. Any differences in early information-gathering strategies may also result in differences in the long-term, with good learners ultimately moving more efficiently [42]. To investigate potential differences, we isolated sections of trajectories where birds were in transit between places of interest, such as foraging patches or resting locations. We then assessed whether performance on a spatial cognition task, measured early in life, was related to changes in straightness and/or speed of their transitory paths after release into the wild several weeks later.

# 2. Methods

## 2.1. Subjects and housing

We hatched 190 chicks (87 females and 103 males) on 25th May 2017 at North Wyke Rothamsted Research farm (Devon, UK, 50°77′ N, −3°9′ W). They were the offspring of adults (16 males and 24 females) that we had caught in the wild and housed in mixed groups for breeding. These mixed groups meant that there were likely to be few full siblings, reducing any potential clutch effects on either cognition or movement behaviour. Actual relatedness between individuals used in this study was calculated from blood samples for another study [43] (electronic supplementary material, Appendix S1). Once laying was completed, we released the adult birds at a location 6.9 km away from the study site. Pheasants rarely disperse more than 3 km from release sites [44], so our chicks would be unlikely to come into contact with their parents or other experienced individuals once released. While we cannot explicitly say that there were no adults near the release site, there were likely few, if any, due to our trapping and relocation regime. The chicks were randomly allocated to one of four indoor enclosures (1 m × 2 m) with replicated environments (perches, drinkers and food bowls) in approximately equal-sized groups (2 × 47, 2 × 48). Chicks were given *ad libitum* access to age-specific commercial chick crumb (Sportsman Game Feed, London, UK) and water. At two weeks old, birds were individually labelled with numbered patagial wing tags (Roxan Ltd, Selkirk, UK).

## 2.2. Assessment of spatial cognitive ability

From one day old, the chicks were trained to associate food with human presence. From five days old, chicks were shaped to enter a testing chamber (75 cm × 75 cm) from their enclosure through a sliding door, first in groups and later individually, to eat mealworms scattered throughout the chamber. Chicks exited the chamber into a 'post-testing' area (0.75 m × 1.25 m) through a pulley-assisted door and were released back into the enclosure once all birds had entered the 'post-testing' area. After two weeks, all chicks would voluntarily enter and exit the testing chamber alone. For the next five weeks, all birds were subjected to an identical series of cognitive tasks unrelated to the current study. Since all prior tasks were the same in design and number of trials, it is unlikely that prior experience influenced performance on the spatial task used for this study.

Maze tasks have previously been shown to be linked to spatial behaviour in the wild [45]. At seven weeks old, we tested the spatial cognitive ability of the pheasant chicks using a radial maze style task (figure 1) in which they had to learn and remember the location of a reward, indicative of their ability to learn about spatial routes and/or landmarks and rely on memory when deciding where to move [31,46]. The birds voluntarily entered the testing chamber alone and were lured to a central platform

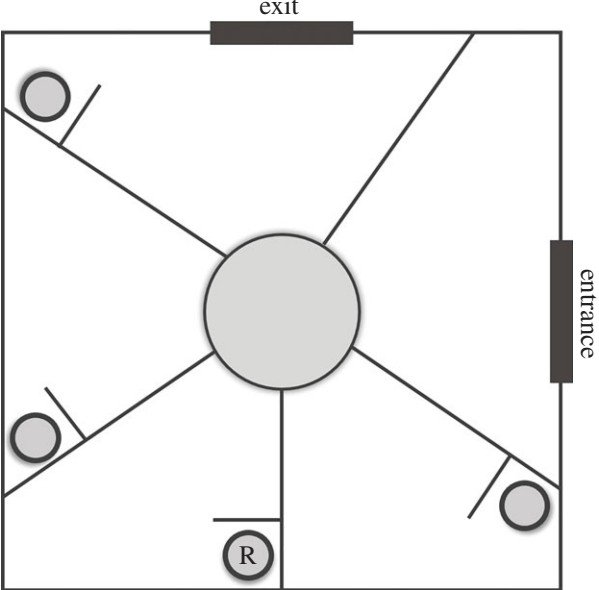

**Figure 1.** Schematic of the spatial task. Birds walk through the entrance and onto the central platform before beginning the task. R denotes the location of the reward (three mealworms).

(20 cm diameter) by a dead mealworm presented in the middle of the platform. Five walls radiated from the central platform, separating the testing chamber into five compartments. At the end of four of the walls (excluding the compartment through which the birds entered the testing chamber), there was a second short wall perpendicular to the main compartment wall that obscured the view of a circular tray. In one of these trays (the location of which was consistent for all birds (figure 1)), we placed a reward of three mealworms, while the remaining trays were empty. Birds were required to walk to the end of a compartment and look around the perpendicular wall to retrieve the reward. Stepping off the central platform into an incorrect compartment (including the entrance arm) was counted as an error. Birds could not move from one compartment to another without returning to the central platform. In each trial, we counted the number of errors a bird made until it retrieved the reward, after which the bird was immediately released from the testing chamber. Birds were also released from the testing chamber, and their trial abandoned, if they appeared stressed (e.g. through lost-calling or flapping). Each bird had the opportunity to complete 12 trials between 3rd and 7th July 2017 and 168 birds completed all 12 trials and could therefore be used in further analyses. The order in which individuals entered the testing chamber was recorded in each trial. Test order has been previously found to be repeatable in pheasants and is thought to indicate motivational traits [47]. We calculated a median test order of a bird over all 12 sessions. Birds with lower median test order could be considered more motivated by food rewards as they were consistently early to enter the testing chamber.

## 2.3. Release

The pheasants were sexed (by plumage) and weighed (Slater Super Samson spring balance – precision 5 g) when 10 weeks old on the 26th July 2017. We fitted birds with tracking tags attached with a backpack harness that comprised elastic wing-straps threaded through heat-shrink tubing. Tags weighed 22 g, which was a mean of 2.63% of released body mass (range = 2.00–3.67%), although birds were expected to continue to grow meaning that after a couple of months, tags were expected to weigh between 1.1% and 2.3% of adult body mass [48]. We released the pheasants into a 4000 m$^2$ enclosure (hereafter the release pen) within a small woodland on North Wyke farm. The release pen was surrounded by a 2 m high wire fence and an exterior 30 cm high electric fence to protect the pheasants from terrestrial predators such as red foxes, *Vulpes vulpes*, while the birds acclimatized to the wild. The release pen contained patches of vegetation providing roosting and shelter sites, as well as *ad libitum* access to water and food from feeders and drinkers. Birds typically remained within the pen for approximately four weeks, but they could voluntarily leave the release pen by flying out and could return either by flying in or walking through one-way holes. Until 30th August, we actively guided birds back into the release pen at dawn and dusk if they had flown over the fence while descending from overnight roost sites or

while moving around the pen during the day. This is a well-established protocol for game-keeping and allows the birds time to acclimate to living in the 'wild', and keeps them safer from terrestrial predators after being released from captivity. This meant that movement outside the release pen was limited until 1st September. From the start of September, birds began to increasingly disperse into the surrounding landscape which consisted of a mix of grassland and woodland and contained 39 more barrel-feeders. There was no game shooting or predator control on the field site during the study.

## 2.4. Monitoring movement

We used a recently developed reverse-GPS system (ATLAS) [36,37] to track the movement of the pheasants. Briefly, this system comprised four time-synchronized, fixed-location receiver stations that surrounded the release site (0.5–4 km away) that recorded the time of arrival of individually identifiable radio signals from tags. The tags on our pheasants emitted a signal once every 4 s. The time of the arrival of signals at each of the receiver stations was used to compute localizations. We then filtered the raw location data (electronic supplementary material, Appendix S2) and computed median locations for each 5 min period. Since pheasants move relatively slowly (less than 0.5 m/s, unpublished data), we felt this was an acceptable reduction in temporal resolution to improve overall accuracy. ATLAS has been reported to have a median accuracy of 5–10 m [36].

Because we were interested in how movements developed, we needed to assess how much of the environment the birds had explored before we stopped actively trying to keep them in the release pen (1st September). However, due to the novelty of the system, we experienced unexpected technical difficulties which meant that our system only started recording locations of tagged birds from 17 August 2017, 22 days post-release. In addition, due to an electronics fault, we had intermittent tag transmission failures throughout the season leading to patchy location data for some dates and individuals and total loss of data for others (electronic supplementary material, Appendix S3). We therefore cannot determine exactly when each bird left the release pen for the first time; however, at a population level, there were very few birds that were tracked away from the release pen between 22nd August and 1st September (electronic supplementary material, Appendix S4). The mean home range size (95%-Kernel density estimate (KDE)) of the pheasants increased over September (1–15 September: mean ± s.d. = 3.97 Ha ± 3.12; 16–30 September: mean ± s.d. = 10.69 Ha ± 12.48), yet the core range (50%- KDE) at the end of the month of still overlapped heavily (mean ± s.d. = 0.54 ± 0.18) with the core range from the beginning of the month (figure 2). Home ranges and overlap were calculated using the *adehabitatHR* R package (v. 0.4.18, [49]). This demonstrates that while the birds are beginning to explore the area, they were not dispersing to completely different areas or moving in a nomadic way. Thus, birds can use their previous experience of an area to inform their movements. We therefore conclude that, although we do not know if a particular individual has been outside the release pen before, we are confident that all birds are at an early stage of exploring the environment outside the pen. We also confirmed (through finding corpses) 39 deaths from predation during the first two months post-release. To ensure that movements were comparable between individuals, we restricted our analysis to movement data collected from live birds with functioning tags in September (electronic supplementary material, Appendix S5). By assessing movement only in September, we obtained a reasonable overview of how the birds develop their movement strategies in an environment with which they are relatively, if not completely, unfamiliar. We only included birds in the analysis that had completed all 12 trials in the spatial task and for which we had obtained daytime localizations for at least 6 h/day for a minimum of seven days in September. This allowed us to consider 50 individuals for subsequent analysis (25 females, 25 males) (total 114 067 locations, mean ± s.d. per individual = 2281.34 ± 774.03).

## 2.5. Behavioural classification

Movement behaviours can be indicative of movement goals [7,50] and movement states such as transit, foraging or resting can be inferred by identifying discrete patterns in turning angles and step lengths (the distance between location estimates). A hidden Markov model (HMM) is a time-series model that is a commonly used and flexible tool for this type of analysis which enables the sequence of state changes to be estimated via the Viterbi algorithm [21,51,52]. We used a HMM from the R package *moveHMM* v. 1.6 [53] on ATLAS data to identify three different movement states, allowing us to analyse transitory paths independently of other types of behaviour. Pheasants are diurnal, therefore we only used paths from between civil dawn and civil dusk (calculated using the *crepuscule* function from the *maptools* package v. 0.9-5 [54]). We split paths with more than a 1 h window of missing points for each individual. This was

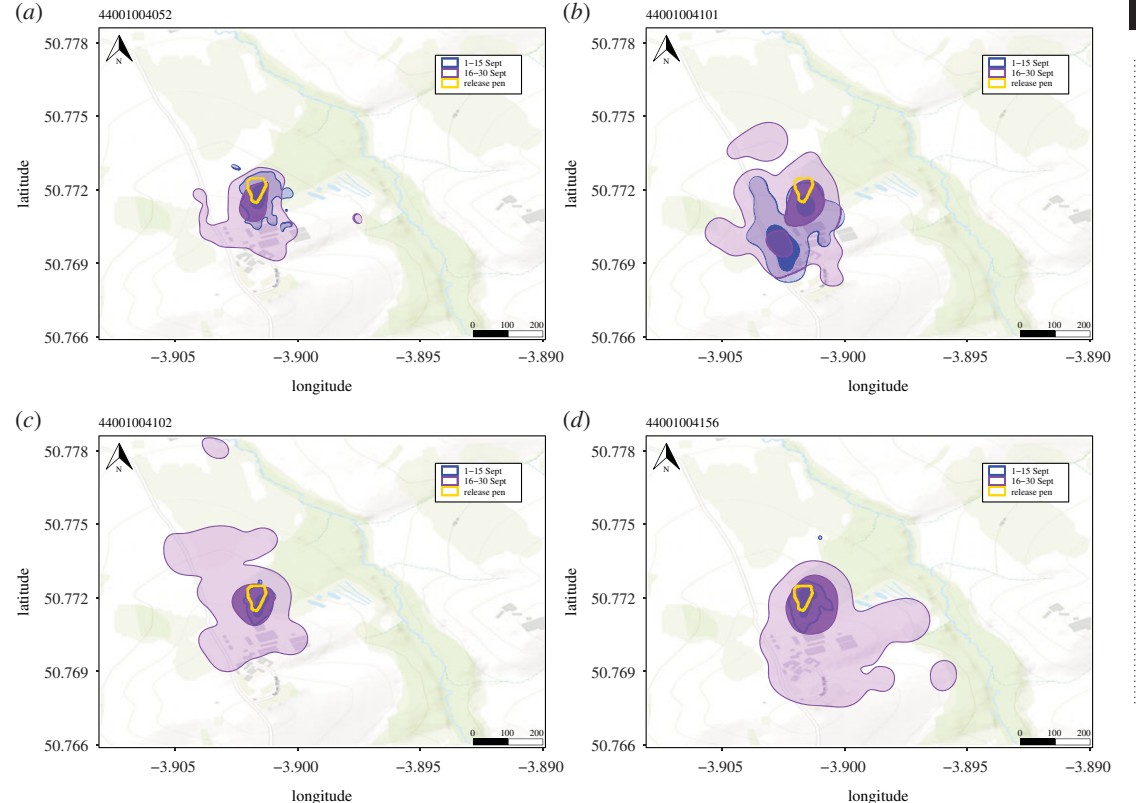

**Figure 2.** Example of how individual home ranges change over the study period. Initially, birds' core home ranges (50% KDE: solid blue (beginning of September)) are very close to the release pen (yellow) and the total home range (95% KDE: transparent colours) is small, although this varies between individuals. At the end of the month, the core home range (slightly transparent purple) still overlaps heavily with the range from the start of the month (demonstrating that they can use their experience to inform their movements) but the total home range (transparent purple) is larger. This indicates that the birds had not formed their final home ranges before the study began, when we were unable to track the birds.

to avoid using interpolation to simulate locations in the missing period which, since pheasants often move slowly, can lead to misclassification of paths as a result of many short steps in a straight line being created. We ran 25 randomizations of the initial parameters for a three-state model to assess model sensitivity and ensure that the model had identified the maximum log-likelihood estimates of the parameters [55]. Initial parameters are described in electronic supplementary material, Appendix S6. Fifteen of the 25 models converged on similar log-likelihood values (within 0.03), indicating numerical stability and good initial parameters. Since some steps were of length zero, we also estimated zero-inflation within the model [53]. We used a Gamma distribution to describe step length and a von Mises distribution for turning angles. The model with the largest maximum log-likelihood separated behaviours primarily by step length (see electronic supplementary material, Appendix S7).

Visual inspection of the HMM-categorized movements enabled us to match the states to their likely behaviours (figure 3). State 1, which we deemed to be resting behaviour, was rare during the day and characterized by essentially no movement (Step length (mean ± s.d.) = 3.504 m ± 2.595 s.d., number of steps = 9961; displacement distance: 7.987 m ± 6.021 s.d.; number of paths = 818). State 2 involved much longer mean step length and displacement distance (Step length (mean ± s.d.) = 33.160 m ± 29.151 s.d., number of steps = 19 674; displacement distance: 93.931 m ± 77.410 s.d.; number of paths = 970) and we deemed this to be transit. State 3 was characterized by an intermediate step length and displacement distance (Step length (mean ± s.d.): 14.240 ± 10.033, number of steps = 84 432; displacement distance: 50.776 m ± 38.540; number of paths = 1834) and frequently occurred in the vicinity of feeding stations, so we called this foraging behaviour. All turning angles were centred around π, which indicates a high frequency of course reversals, although to a lesser degree in transit behaviour. We expect that this is due to slow movement speeds coupled with some error which may cause the paths to look as if a bird is moving backwards and forwards when in reality it has moved only slightly or is still. It could also be indicative of pheasants moving around obstacles. We used the Viterbi algorithm (*viterbi* function in

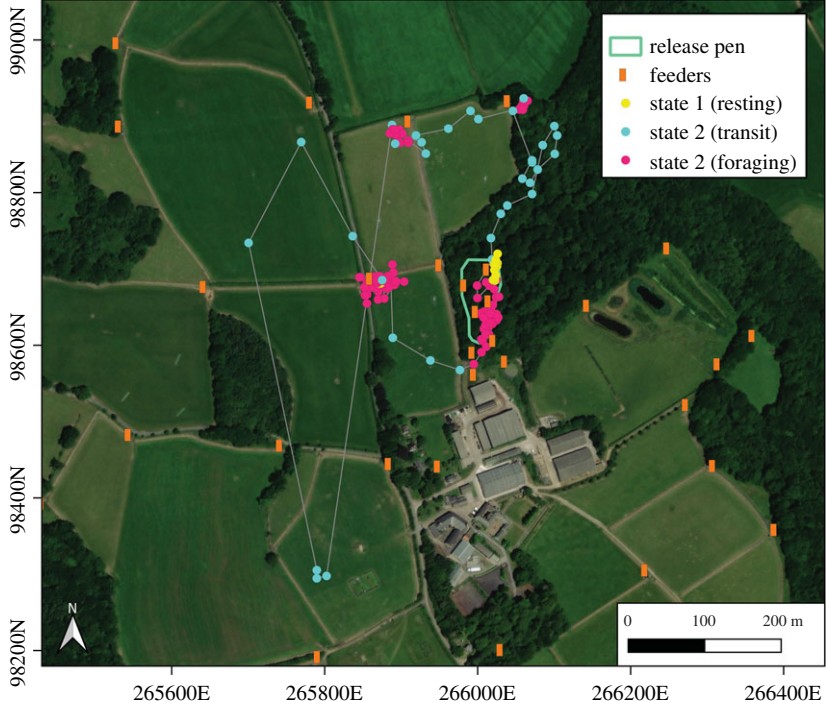

**Figure 3.** Example HMM classification of a pheasant track on 21st Sept 2017. Movements start from resting state, briefly foraging in the pen before leaving in a northward direction. The bird forages near two feeders before quickly moving south (perhaps after being startled by a predator) then making its way back towards another feeder and returning to the woodland where it forages at the edges of the release pen, where there is a lot of leaf litter.

*moveHMM*) to estimate the most likely sequence of state changes for each trajectory based on the best-fitted model and isolated the sections of the trajectories that were determined to be 'transit'.

## 2.6. Assessing search efficiency

We only considered movement during transit (State 2) to be indicative of demands on efficiency. The mean displacement distance during a transit bout (absolute distance moved during that movement bout) was about double the displacement distance covered during a foraging bout (State 3) and usually occurred between foraging bouts, indicating that the focal bird was moving in unprofitable land between profitable foraging patches. It is this movement between foraging sites that could be made more efficient with experience. We consider more efficient paths to be straighter and/or faster to travel. Paths may be straight and/or quick simply because an individual has learned the shortest route to their goal. Alternatively, the time that an individual takes to move between two distant points may be determined by their ability to select terrain that is easier to traverse, for instance flat ground or roads. Considering both straightness (straight line distance/path length) and speed (m/s) therefore provides us with two metrics of movement efficiency through both turn frequency and the relative ease of the terrain. We isolated each section of all paths that were designated as transit (State 2) and assessed both straightness and speed using the *amt* package [56]. Using this approach, we can objectively identify differences in the characteristics of transitory paths over time, without the need to pre-determine the goal locations e.g. feeders. This is important because the birds may be travelling to other types of foraging sites e.g. leaf litter to forage for arthropods, or to refuges. The *straightness* function was used to calculate straightness and we calculated speed by measuring the total distance travelled (*tot_dist* function) divided by duration in seconds. We were concerned that paths that were very long in duration may have been misclassified. The mean (±s.d.) duration of transitory movements was 1.7 h (±1.93) but the maximum value was 13.5 h which we do not feel is likely from a biological standpoint. We therefore removed transitory paths that were over 6 h in duration (95th percentile = 5.68 h, $n = 45$). In total, we analysed the 925 transitory paths (mean number of steps in path = 16.14 [range = 3–69]) that were shorter than 6 h in duration that were made by birds that completed all trials for the cognition task.

## 2.7. Statistical analyses

A criterion based approach to assessing memory (e.g. trial number at which 10 consecutive correct response trials are completed [57]) is a popular and effective measure, but requires the completion of many trials, sometimes hundreds. Due to logistical and temporal constraints associated with assaying high numbers of young pheasants, we used a 'reverse criterion' approach, considering the maximum number of consecutive trials completed with no errors as a measure of accuracy, with birds with higher scores being assumed to have the better spatial cognitive ability. This measure represents the stage of learning the bird had achieved at trial 12, with birds which had not learned the task well only completing very few no-error trials consecutively, perhaps by chance. Alternatively, birds successfully completing a high number of consecutive trials are unlikely to have achieved this by chance ($p = 0.2$ per trial). We used a Chi-squared test to assess whether the distribution of observed scores was different from that expected by chance. *P*-values were calculated using Monte Carlo simulations ($n = 2000$) since some of the expected frequencies were low.

To identify whether any cognitive or non-cognitive traits influenced the speed or straightness of transit paths, we conducted two GLMMs. Any continuous variables were centred and scaled using the *scale* function. We compared different error structures by plotting the residual and fitted heteroscedasticity of different models and found that a Gamma error structure with a log link function provided the best fit for the data for both models. We included a two-way interaction between date (as a scaled integer) and performance on the spatial task to identify changes over time attributed to spatial cognitive ability. We hypothesized that food-motivation may play a role in movement between patches where individuals exhibit clearer goal-directed behaviour, perhaps faster or more direct routes. We therefore included scaled test order, as well as sex as fixed effects in the full model. In the straightness model, we also included the number of locations within the path (scaled) as a fixed effect. This was not included in the speed model as this is already part of the speed calculation (since number of locations is colinear with time). Bird identity was included as a random effect in both models. We reduced the model using stepwise model simplification: optimal models were selected based on AIC values and residual variance, calculated from likelihood ratio tests (base R function: *drop1*, test = Chi). All analyses were performed in R (v. 3.5.3) [58] using the R Studio wrapper (v. 1.2.1335) [59].

## 2.8. Ethical considerations

Handling of all pheasants during rearing and testing was kept to a minimum. Task participation was voluntary and only positive reinforcement was used. Birds were habituated to experimenters and the testing chamber from their first day of life to help alleviate the stress that testing procedures may cause. Chicks were kept in less densely populated conditions than is recommended by DEFRA's code of practice [60]. When capturing wild adults for breeding, traps were checked at least three times per day. All work was conducted under Home Office license PPL 30/3204 and approved by the University of Exeter Animal Welfare Ethical Review Board.

# 3. Results

## 3.1. Did birds learn the spatial task?

The 50 birds used in the movement analysis varied in performance score (maximum number of consecutive no error trials) from a minimum of 0 to a maximum score of 9 (figure 4) (distribution of all 168 birds' scores can be seen in electronic supplementary material, Appendix S8). The distribution of observed scores was different to that expected by chance ($\chi^2 = 14213$, n Monte Carlo simulations = 2000, $p < 0.001$; figure 4).

## 3.2. Association of spatial cognitive ability on movement behaviour

Individuals that had better performance scores in the spatial task improved the speed of their transitory paths more rapidly over the month than poor learners, indicated by a significant interaction between date and accuracy score (table 1). However, better performers also started with slower paths and eventually improved over time to attain the same level as worse performers (figure 5a). Performance on the

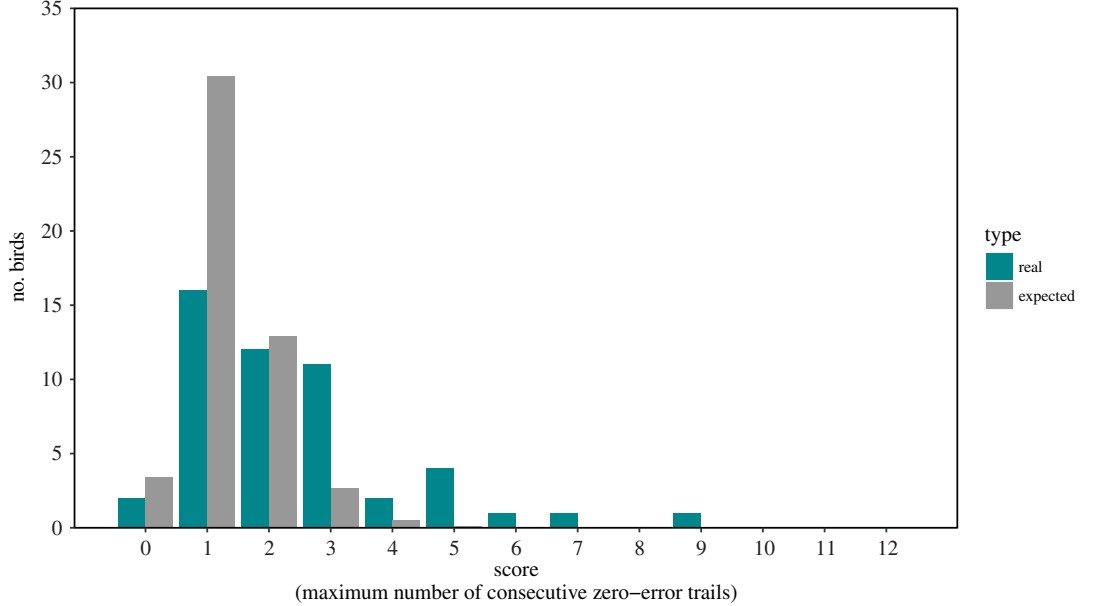

**Figure 4.** Distribution of observed scores on the spatial cognition task of pheasants in captivity (green) and expected values (grey).

**Table 1.** Model outputs from generalized linear mixed model of the effects of performance in the spatial task on the speed and straightness of transitory paths. $P$ values and likelihood ratio test values are given when they were removed from the full model, denoted by superscript.

| parameter | $\beta \pm$ SE | LRT | $p$ |
|---|---|---|---|
| speed | | | |
| sex (M)[1] | $-0.050 \pm 0.074$ | 0.482 | 0.488 |
| test Order[2] | $0.051 \pm 0.035$ | 2.059 | 0.151 |
| date* | $0.045 \pm 0.050$ | — | — |
| performance* | $-0.014 \pm 0.019$ | — | — |
| date $\times$ performance* | $0.034 \pm 0.016$ | 4.460 | *0.034* |
| straightness | | | |
| test order[1] | $-0.002 \pm 0.039$ | 0.05 | 0.966 |
| sex[2] | $-0.017 \pm 0.078$ | 0.05 | 0.824 |
| date $\times$ performance[3] | $0.004 \pm 0.013$ | 0.08 | 0.781 |
| performance[4] | $0.004 \pm 0.021$ | 0.05 | 0.825 |
| date* | $0.102 \pm 0.024$ | 17.31 | *<0.001* |
| number of locations in path* | $-0.514 \pm 0.022$ | 372.09 | *<0.001* |

*denotes the terms are present in the reduced model.

spatial cognition task was not related to the straightness of transitory paths but overall, birds increased the straightness of their paths over time (figure 5*b*). We found no differences in overall movement efficiency based on an individual's sex or motivation (table 1).

## 4. Discussion

Pheasant chicks improved their performance in a five-arm radial maze task, indicative of their ability to acquire spatial information and use spatial memory but individuals varied in their task performance. We then released these birds into a novel real-world environment and tracked their movements to investigate

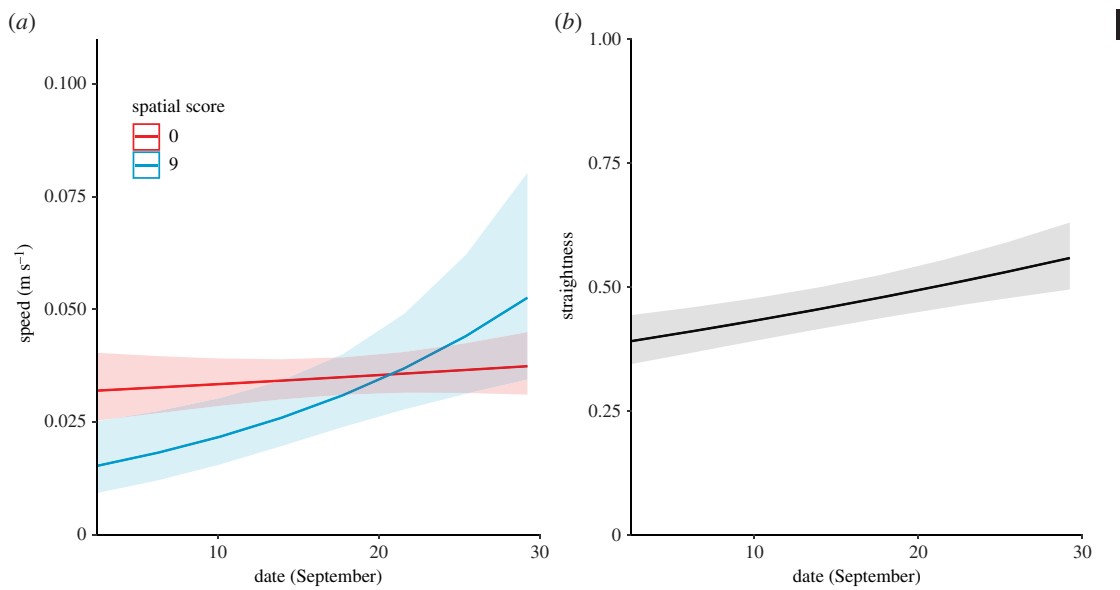

**Figure 5.** Predicted values of (*a*) speed for low performance (0 = red) or high performance (9 = blue) on the spatial task. Increase in straightness (*b*) is shown over all individuals. We found no effect of spatial cognitive ability therefore this was not present in the minimal model (shown). Model includes scaled day (integer of date) but x-axis has been transformed for clarity. Shaded areas represent 95% confidence intervals.

whether they became more efficient in their transitory paths as they gained experience of the environment. Due to technical difficulties we were unable to track birds from their initial entry into the novel environment. However, at this point, few birds were roaming outside the release pen and the majority of birds remained close to the release pen until mid-September, indicating that they only began exploring the wider environment in September. We found that individuals that were better performers in our test of spatial cognitive ability early in life initially made slower transitory paths than worse performers at the beginning of the study period. However, over a period of one month, these better performers improved their transitory path speeds more rapidly, to reach speeds similar to those of worse performers. All birds, regardless of performance on the cognitive task, increased the straightness of their transitory paths throughout the period, suggesting that they were learning about the environment and optimal routes between foraging sites. This study provides the first empirical insight into how inherent individual differences in spatial cognitive ability might be linked to the development of transitory paths in free-roaming animals.

Individuals that performed well in the spatial task moved more slowly than poor performers at the beginning of the study period. Individuals that performed poorly on the task exhibited a fairly consistent speed and the better performers increased their speed to match the speed of poorer performers by the end of the study. Indeed, their slopes of improvement suggest (figure 5*a*) that better performers may have eventually moved faster than poor performers. These differences could not be explained by differential opportunities to learn because all birds were initially naïve to the environment and, being released on the same day at the same site, had equal experience of it. It is also unlikely to be explained by differential access to informed individuals as we had removed all parents from the study site and did not detect them on the site again during our tracking period. Although there is the possibility of our test birds following wild, experienced non-parent adults that we did not capture or detect, all released birds had equal access to them. With this being said, since we were unable to track the birds for the first month of release, it is possible that a few birds became more experienced than others and that these individuals could be followed. However, if this was the case then we would expect birds to move at similar speeds, independent of their score on the cognitive task. Since we find differences in speed are associated with spatial cognitive ability, the birds are likely to be following individuals of similar spatial ability or are moving independently of other birds. In random search processes, speed, turning patterns and perception govern the success of encounters with resources [26]. Slower movements by better-performing birds during the early occupation of a novel landscape may facilitate the gathering of more information at this early period and could indicate that a speed-accuracy trade-off is involved in early exploration in novel environments [38]. Since better performers

did not use faster or straighter paths than poor performers by the end of our study period, we were unable to detect a clear advantage in movement efficiency as a consequence of these early differences.

In accordance with other studies [21,22], we found that pheasants in general increase the straightness of their transitory paths as they gain experience of an environment. However, in contrast with our predictions and our results considering path speed, we found no difference between individuals of differing spatial cognitive ability in the straightness of their transitory paths. The lack of relationship between spatial cognitive ability and straightness of transitory paths could have several explanations. First, in species where transit can occur via direct routes with few obstacles, straightness is an obvious and informative measure of efficiency since turning is costly [23]. In the case of a terrestrial bird in a rural landscape, the presence of obstacles and varying terrain may mean that the most 'efficient' path is not necessarily the straightest, with costs of movement or risk being higher for some apparently direct paths. This measure becomes further complicated when we consider multiple start and end points of transitory paths both between and within individuals over different days. For example, birds may be moving towards any of the 39 feeders at our site, and the most efficient route in open, flat grassland may be markedly different in straightness to that in woodland. Such a measure becomes further complicated when we consider not just energetic costs, but also those of exposure to risk, with the safest routes between two points including a detour to avoid a risky site. Predator risk and social factors are also likely to be highly fluctuating and unpredictable and could also influence the end point of a transitory path [61–63] and social factors in particular may influence how an individual moves if part of a group [64,65]. Furthermore, foraging strategy or diet preferences may differ between individuals, leading to differences in movement strategies [4,66]. Therefore, although all pheasants showed an increase in path straightness over the study period, indicating improved route choice with experience, it is perhaps not surprising that we did not find individual differences that related to spatial cognitive ability; especially given the likely high level of noise in the measures of straightness across routes and the conditions under which those routes are travelled.

Our findings appeared to be robust to differences in sex and motivation as we found no effects of these factors on performance on the spatial task, movement speed or straightness in our study. Previous research has found distinct sex differences in spatial cognitive ability linked to larger home ranges [67], more complex habitats [68] or differential breeding biology [69] for one particular sex, although see [70]. Pheasants display pronounced sex differences in movement and space use, exhibiting sexual segregation between November and February [35] and with females typically dispersing further than males [71]. It is therefore surprising that we did not find differences between the sexes in spatial cognitive ability and/or movement efficiency. However, failure to detect a sex difference may be due to our short study period focussed early in the bird's life and annual cycle, i.e. before males have established territories and before females began to search for mates.

The proposed relationship between spatial cognition and movement in the real world, presumed to be key to efficient space use [2,7], has lacked empirical support. We have used the widely accepted method of measuring performances on abstract cognitive tasks to provide a useful assay of inter-individual variation in cognitive abilities [72,73] and we have linked this to changes in movement behaviour, specifically changes in speed, in natural, real-world landscapes. This is reassuring in that it links performance in controlled tests to a natural behaviour that is assumed to involve the cognitive process being tested. Such links between tasks and natural behaviours are rare (but see [18,74]). While it may be appealing to simply think that 'better' spatial memory, as assayed by our task, should bring intuitive fitness benefits in terms of more efficient travel between desired locations (perhaps through better survival or reduced energy expenditure), we were unable to find evidence for the quicker or straighter movements that we expected from birds with 'good' spatial memory. Instead, we found that better performers made slower movements in the early exploration of an environment. However, any long-term benefits arising from these early differences in movement strategies remain to be explored.

Data accessibility. Data and relevant code for this research work are stored in GitHub: https://github.com/CBeardsworth/pheasant_spacog_movement and have been archived within the Zenodo repository: https://doi.org/10.5281/zenodo.4516391.

Authors' contributions. C.E.B. and J.R.M. conceived the idea for the manuscript; C.E.B. conducted the analyses and led the writing of the manuscript; C.E.B., L.A.C., P.R.L., E.J.G.L., M.A.W., J.O.v.H. and J.R.M. collected the cognition data; C.E.B., M.A.W. and J.R.M. collected the movement data. R.N., Y.O. and S.T. developed the reverse-GPS system and provided support throughout data collection. All authors contributed critically to the drafts and gave final approval for publication.

Competing interests. We declare we have no competing interests.

Funding. The empirical work was funded by an ERC Consolidator Award (616474) awarded to J.R.M. The development of the ATLAS system was possible through long-term funding from the Minerva Center for Movement Ecology and the Minerva Foundation. ST was supported by grants 965/15, 863/15, and 1919/19 from the Israel Science Foundation.

Acknowledgements. We thank Rothamsted Research North Wyke for accommodating the rearing and release of the pheasants. We also thank Dr Camille Troisi for her help with data collection and animal husbandry. We are grateful to the long-term team members at the Minerva Center for Movement Ecology for their persistence in developing ATLAS. Thank you also to Dr Andrew Higginson and Dr Chloe Stevens for their comments on previous drafts of the manuscript and to Sergio Olivera for his help with statistics. We also thank three anonymous reviewers for their useful and detailed comments on the manuscript.

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
