## [Peer Review File · Royal Society Open Science]

Review History

RSOS-201758.R0 (Original submission)

Review form: Reviewer 1

Is the manuscript scientifically sound in its present form?

No

Are the interpretations and conclusions justified by the results?

No

Is the language acceptable?

Yes

Do you have any ethical concerns with this paper?

No

Reports © 2021 The Reviewers; Decision Letters © 2021 The Reviewers and Editors; Responses © 2021 The Reviewers, Editors and Authors. Published by the Royal Society under the terms of the Creative Commons Attribution License <http://creativecommons.org/licenses/by/4.0/>, which permits unrestricted use, provided the original author and source are credited

Have you any concerns about statistical analyses in this paper?

No

Recommendation?

Reject

Comments to the Author(s)

Please see attached file (Appendix A).

Review form: Reviewer 2

Is the manuscript scientifically sound in its present form?

Yes

Are the interpretations and conclusions justified by the results?

Yes

Is the language acceptable?

Yes

Do you have any ethical concerns with this paper?

No

Have you any concerns about statistical analyses in this paper?

No

Recommendation?

Accept as is

Comments to the Author(s)

The authors have put a substantial amount of work into revising this manuscript and it has paid off.

Decision letter (RSOS-201758.R0)

Dear Ms Beardsworth,

The Editors assigned to your paper RSOS-201758 "Spatial cognitive ability is associated with speed of movement in the early exploration of an environment" have now received comments from reviewers and would like you to revise the paper in accordance with the reviewer comments and any comments from the Editors. Please note this decision does not guarantee eventual acceptance.

Please submit your revised manuscript and required files (see below) no later than 21 days from today's (ie 09-Nov-2020) date. Note: the ScholarOne system will 'lock' if submission of the revision is attempted 21 or more days after the deadline. If you do not think you will be able to meet this deadline please contact the editorial office immediately.

on behalf of Dr Claudia Wascher (Associate Editor) and Kevin Padian (Subject Editor)
openscience@royalsociety.org

Subject Editor Comments to Author:

We have two further reviews for your manuscript. A previous reviewer feels that you have answered concerns sufficiently, but a second does not and independently came up with the same concerns that other previous reviewers had and that do not seem to have been sufficiently addressed. Please take particular care in your revision because we now have at least two reviewers who are voicing the same concerns, and they should be addressed. As is our policy, if reviewers are not satisfied with the next revision we will be unable to consider the manuscript further. Best wishes and thanks for submitting to RSOS.

Associate Editor Comments to Author (Dr Claudia Wascher):

The presented study investigates the relationship between performance in a spatial memory tasks and movement patterns after release in young pheasants. Both reviewers find the experiment well designed. I was able to secure one of the original reviewers, who complements the authors on the substantial amount of work they have invested in the revision of their manuscript. The second reviewer voices concerns regarding the framing of the study as developmental, given that the authors have no data on movement for a number of days after release due to technical issues.

I agree that the framing of the study should be adapted to the actual data the authors present. I am looking forward to read the revised version of the manuscript.

Reviewer comments to Author:

Reviewer: 1

Comments to the Author(s)

Please see attached file.

Reviewer: 2

Comments to the Author(s)

The authors have put a substantial amount of work into revising this manuscript and it has paid off.

===PREPARING YOUR MANUSCRIPT===

===PREPARING YOUR REVISION IN SCHOLARONE===

Author's Response to Decision Letter for (RSOS-201758.R0)

See Appendix B.

RSOS-201758.R1 (Revision)

Review form: Reviewer 1

Is the manuscript scientifically sound in its present form?

Yes

Are the interpretations and conclusions justified by the results?

No

Is the language acceptable?

Yes

Do you have any ethical concerns with this paper?

No

Have you any concerns about statistical analyses in this paper?

No

Recommendation?

Major revision is needed (please make suggestions in comments)

Comments to the Author(s)

See attached file (Appendix C).

Decision letter (RSOS-201758.R1)

This year has been very difficult for everyone, and we want to take the opportunity to thank you for your continued support in 2020.

The Royal Society Open Science editorial office will be closed from the evening of Friday 18 December 2020 until Monday 4 January 2021. We will not be responding during this time. If you have received a deadline within this time period, please contact us as soon as possible to allow us to extend the deadline. If you receive any automated messages during this time asking you to meet a deadline, we offer apologies and invite you to respond after the festive period or during normal working hours.

With our best for a peaceful festive period and New Year, and we look forward to working with you in 2021.

Dear Ms Beardsworth

The Editors assigned to your paper RSOS-201758.R1 "Spatial cognitive ability is associated with speed of movement in the early exploration of an environment" have now received comments from reviewers and would like you to revise the paper in accordance with the reviewer comments and any comments from the Editors. Please note this decision does not guarantee eventual acceptance.

Please submit your revised manuscript and required files (see below) no later than 21 days from today's (ie 21-Dec-2020) date. Note: the ScholarOne system will 'lock' if submission of the revision is attempted 21 or more days after the deadline. If you do not think you will be able to meet this deadline please contact the editorial office immediately.

on behalf of Dr Claudia Wascher (Associate Editor) and Kevin Padian (Subject Editor)
openscience@royalsociety.org

Associate Editor Comments to Author (Dr Claudia Wascher):

Associate Editor: 1

Comments to the Author:

The authors have clarified some aspects of the manuscript, namely the justification of the developmental aspects of movement patterns. The reviewer still raises concerns whether the provided data and analysis sufficiently support the conclusion. Questions arise whether the relationship between movement patterns in the wild and performance in a spatial cognitive task reflect learning abilities of individuals or whether alternative factors, for example, boldness could explain the described patterns.

Associate Editor: 2

Comments to the Author:

(There are no comments.)

Reviewer comments to Author:

Reviewer: 1

Comments to the Author(s)

See attached file.

===PREPARING YOUR MANUSCRIPT===

===PREPARING YOUR REVISION IN SCHOLARONE===

<https://royalsociety.org/journals/authors/author-guidelines/#supplementary-material> to include a suitable title and informative caption. An example of appropriate titling and captioning may be found at https://figshare.com/articles/Table_S2_from_Is_there_a_trade-off_between_peak_performance_and_performance_breadth_across_temperatures_for_aerobic_sc_ope_in_teleost_fishes_/3843624.

Author's Response to Decision Letter for (RSOS-201758.R1)

See Appendix D.

Decision letter (RSOS-201758.R2)

Dear Ms Beardsworth,

It is a pleasure to accept your manuscript entitled "Spatial cognitive ability is associated with transitory movement speed but not straightness during the early stages of exploration" in its current form for publication in Royal Society Open Science.

on behalf of Dr Claudia Wascher (Associate Editor) and Kevin Padian (Subject Editor)
openscience@royalsociety.org

Appendix A

RSOS Review – 27 Oct 2020

Found this manuscript to be an Interesting addition to the literature, I think it is an important step to link spatial cognitive ability to the development of movement paths. I additionally find the experimental design to be well done.

My main concern comes from the author framing the analyzed movement paths as indicative of spatial learning because movement data were not analyzed for the first 36 days after release. That is a long time for individuals to become habituated to and learn about the environment, likely obscuring many differences among individuals in learning that influenced the rate of change of speed and path straightness. Nevertheless, the authors find individuals with better spatial memory were initially slow in their movements but improved over the study period. I think this is a valid contribution to the literature, but the lack of initial data means it is impossible to infer an overall connection between development of movement paths and spatial learning ability. Since there are no data for the first 36 days of movement in the release pen, the analyzed movement data are not representative of the “development of movement paths” that the framing of this paper focuses on. For example, in the 36 days of movement not analyzed, poor performers could have large increases (or decreases) in speed such that what we’re seeing here is the equilibrium after learning the environment. Therefore, I think the authors need to scale back the framing of the paper as the influence of spatial memory on movement *development*.

My second suggestion would be that the authors could augment these data by including social and ecological factors, and potentially by quantifying exploratory traits. Right now, the reader gets a sense that it is unclear what is actually influencing the way the birds are moving around the release pen (i.e. because in the presentation of the data the authors never connect transit phases to the end goal of getting to feeders or drinkers – which is necessary for the assumption that the birds need to learn to be efficient in movement; there is no indication that the birds are moving between the same areas, which is another key prediction if spatial learning occurs - increase in speed over time could result from maturing muscles otherwise; because there are some excessively long transit times present in the data that seem ecologically impossible; because the authors dedicate a portion of the intro to exploration, but do not measure it or bring it up in the discussion). By linking movement data to habitat characteristics, the simultaneous movement of peers, and the personality trait exploration, the authors may find more clear differences among movement of low and high performers on the spatial memory task. I make detailed suggestions about what I mean for these in the line edits.

Lines 69-82 – This paragraph about exploration is a little out of place. I think the ms could benefit from increased clarity on how exploration is different from spatial learning and how that will be reflected in movement paths. Exploration should describe the initial movements of individuals in novel areas, while spatial memory will describe the subsequent movements in familiar areas. Instead of clarifying how individuals use spatial memory to inform movement decisions (the framing of the paper), this paragraph emphasizes how personality traits affect

movement decisions differently under different contexts (which is not relevant to the data presented here because you don't measure exploration).

Lines 105-108 – I appreciate the authors acknowledgment of potential effects of testing behavior in captivity, and I think they justify well the reason that captive individuals are appropriate here.

Line 156 – “number *of* trials”

Figure 1: Exploration could also be measured as the number of visits to incorrect compartments after demonstrating that it has learned the food holding compartment by visiting it at least twice consecutively? i.e. continuing to sample the environment. Although it is unclear whether this occurred.

Lines 200-201 – Exploration could be measured as the latency from release to first leave the release pen?

Lines 205-207 – Can you dumb this down a little bit? Do you mean that you can triangulate the location of the bird from the time it takes the signal to reach each of the base stations during each 5-minn increment?

Lines 220-222 – Where do these data come from? Visual observations? The ATLAS tags? This could also be a measure of individual exploration propensity.

Lines 227-228 – Given how fast the 12 spatial learning trials took in captivity I do think this a significant weakness of the study because spatial learning might have occurred and reached an asymptote already by 37 days post release.

Line 251 – I think it would be good to include the model comparison table in the main text. For example, how are you deciding “best”? Change in AIC? That is important for the reader to interpret the results.

Line 253-254 – This sentence lacks clarity. The first part of the sentence implies that classifications are inferred from inspecting the movement patterns. So, I'm not sure I'm understanding what the second part of the sentence is trying to convey. Unless I am misinterpreting your intended message, I suggest rewording to simply say “We inspected individual movement patterns to categorize each state.”

Lines 263-264 – I think the main text of the ms could benefit from the addition of a figure here to illustrate a typical movement path representative of each state. The relevance of turning angles centered around π is difficult to visualize.

Line 277 – Are profitable foraging patches the feeders in the release pen? I think it would make sense to tie transit paths directly to the end goal of a feeder or drinker, so that it is clear the movement is goal directed and thus benefitted by efficiency.

Line 291 – Even 6 hours in duration seems like a long time to be in one continuous transit phase. Since the mean is 1.7hr, what is your justification for this cutoff?

Line 297 – This citation is not in number form, FYI.

Lines 311-312 – Why did the authors dichotomize here? I think more explanation is needed. Otherwise, the comparison of the observed mean number of consecutively correct trials to the distribution of the simulated mean number of consecutively correct trials seems a more intuitive way to get at this question (e.g. was the observed mean higher than expected by chance, indicating birds as a group were not performing randomly).

Line 317-318 – was the Gamma error structure a “good” fit, or just the best of the poor fit options?

Line 321-322 – Could the authors add a description of why this variable is important to include?

Lines 371-373 – I don’t really find this a convincing argument that the birds were still developing spatial memory. Maybe if there was more information provided about the location of the individuals’ transit paths. For example, are any of the transit paths from outside of the release pen? If so, then I am more convinced that change in straightness and speed of movement reflect some spatial learning. If transit paths are all within the release pen (which is where the base stations are, yes?) then transit paths at the point analyzed could just represent individual differences in activity level.

Line 390 – But peers can be informed individuals if they discover the feeding stations first. Do you have any evidence for or against social information use among the released birds? Is it possible that there was no difference in transit speed of poor performers because they were following a knowledgeable peer that quickly explored the environment and discovered the feeders before the ATLAS tags came online?

Lines 416-417 – Is it possible for the authors to incorporate satellite imagery to assess factors affecting path selection with step selection functions (i.e.

<https://link.springer.com/article/10.1007/s10980-019-00777-z>

)? This is a direct way to address this piece of the discussion.

Line 419 – Are the authors able to use the time + location data to determine if certain individuals move as a group? This could inform an alternative explanation for the results of speed and straightness of movement. There is significant literature indicating that social context influences movement, and that individual differences influence the way in which social

context changes movement (<https://pubmed.ncbi.nlm.nih.gov/24047530/> as one example off the top of my head).

Line 450 – There was no effect of cognitive performance on path straightness so the claim that better performers have “more tortuous movement” is not supported by the data.

Appendix B

Found this manuscript to be an interesting addition to the literature, I think it is an important step to link spatial cognitive ability to the development of movement paths. I additionally find the experimental design to be well done.

We are pleased that the reviewer thinks that the article will be an interesting addition to the literature.

My main concern comes from the author framing the analyzed movement paths as indicative of spatial learning because movement data were not analyzed for the first 36 days after release. That is a long time for individuals to become habituated to and learn about the environment, likely obscuring many differences among individuals in learning that influenced the rate of change of speed and path straightness. Nevertheless, the authors find individuals with better spatial memory were initially slow in their movements but improved over the study period. I think this is a valid contribution to the literature, but the lack of initial data means it is impossible to infer an overall connection between development of movement paths and spatial learning ability. Since there are no data for the first 36 days of movement in the release pen, the analyzed movement data are not representative of the “development of movement paths” that the framing of this paper focuses on. For example, in the 36 days of movement not analyzed, poor performers could have large increases (or decreases) in speed such that what we’re seeing here is the equilibrium after learning the environment. Therefore, I think the authors need to scale back the framing of the paper as the influence of spatial memory on movement *development*.

The reasons why we believe that the birds are largely novel to the environment even after 36 days after release were not very clear in the original MS. We apologise for this and will explain our reasoning and the changes we have made to make this clearer here.

We put the birds in a release pen and attempted to keep them in the release pen by actively walking around the pen at dawn and dusk to usher the birds through small one-way gates in the fences. We did this until 30th August and have tried to explain this in more detail on (L196-201;L216-218). This is standard procedure for released pheasants to keep them safe from predators while they acclimate to the wild. Once we stopped guiding the birds back

into the release pen, we see an increase in dispersal and movement away from the release pen. We show this in the supplementary material (S4) and mention that birds rarely move away from the pen in before September on L224-225 & L382-383. We only analysed data for September, when birds were starting to explore outside the release pen, rather than analysing movement only within the release pen. Importantly, our tracking system covers the release pen and the surrounding area, the receiver stations are 1-4km away and can track birds within the array. We aimed to be completely transparent with our methods as the lack of tracking in the first month inhibited us from being able to prove that movements outside of the release pen were the birds' first ever. However, we are confident that the majority of the birds have done very little exploring outside the release pen before September. Our title infers that this is the early exploration of the environment (not that the environment is completely novel to them, we also reiterate this in L227 & L232) and we believe this is true. Furthermore, it is expected that as birds gain experience, their transitory paths will become straighter and we see that this is the case with the pheasants. We therefore think that our analysis is a true representation of how movement paths develop as birds gain experience in an environment and connect this to cognitive abilities. We feel that framing the paper as the development of movement is therefore accurate based on the data that we have collected and we hope that the reviewer now agrees.

My second suggestion would be that the authors could augment these data by including social and ecological factors, and potentially by quantifying exploratory traits. Right now, the reader gets a sense that it is unclear what is actually influencing the way the birds are moving around the release pen (i.e. because in the presentation of the data the authors never connect transit phases to the end goal of getting to feeders or drinkers – which is necessary for the assumption that the birds need to learn to be efficient in movement; there is no indication that the birds are moving between the same areas, which is another key prediction if spatial learning occurs – increase in speed over time could result from maturing muscles otherwise; because there are some excessively long transit times present in the data that seem ecologically impossible; because the authors dedicate a portion of the intro to exploration, but do not measure it or bring it up in the discussion). By linking movement data to habitat characteristics, the simultaneous movement of peers, and the personality trait exploration, the authors may find more clear differences among movement of low and

high performers on the spatial memory task. I make detailed suggestions about what I mean for these in the line edits.

The suggestion to link transit phases to an end goal is a great idea and one that we have carefully considered in the past. We came to the conclusion that we cannot standardise starting location or goal location in this study as the birds are free roaming. We considered using the feeders as goal locations, but we cannot assume that they always want to arrive at a feeder. These birds also feed on insects, crops, and berries as well as the supplementary food, therefore assuming that their goal is a feeder would significantly limit our inferences. In response to a previous reviewer with a similar comment, we also attempted to use several methods to identify 'goal locations' (e.g. using unsupervised machine learning) but these were unsuccessful. Regardless, the current approach using HMMs allows us to objectively analyse the transitory routes between any places of interest. These places of interest incorporate any place where the birds are either resting or foraging and we believe this is a more effective approach than subjectively determining goal locations for free ranging animals. We have added an explanation between L296-299 to try and make this reasoning clearer and hope that this may alleviate any concerns that future readers may have.

In response to the suggestion of including social or environmental factors in our model, we agree that this is an appealing prospect. The environment, both biotic and abiotic are likely key factors that affect movement in some way. However, while we would like to incorporate them into this paper, they are not trivial to include. This manuscript is already quite long and we feel that each of the environmental and social factors would require a separate paper to fully incorporate their effects on movement. Here, we present one of the first steps to understanding how differences between individuals in solving a spatial task can influence movement in the wild. While we do not cover everything that could influence movement in this paper, we still feel that our contribution to the literature is interesting and useful.

Lines 69-82 – This paragraph about exploration is a little out of place. I think the ms could benefit from increased clarity on how exploration is different from spatial learning and how that will be reflected in movement paths. Exploration should describe the initial movements

of individuals in novel areas, while spatial memory will describe the subsequent movements in familiar areas. Instead of clarifying how individuals use spatial memory to inform movement decisions (the framing of the paper), this paragraph emphasizes how personality traits affect movement decisions differently under different contexts (which is not relevant to the data presented here because you don't measure exploration).

We wrote this paragraph to explain speed-accuracy trade-offs, which comprise a lot of our discussion. We did not intend for it to be understood to be solely commenting on differences in personality traits but instead, a description of the ways animals can collect information and how that can be related to the speed of movement. We reiterate the speed of movement multiple times in this paragraph and therefore feel it is relevant to the MS. To try to prevent confusion for future readers, we have removed "Strategies of spatial exploration" from L69 and replaced it with "Movement strategies" to try and remove the strong association with personality studies.

Lines 105-108 – I appreciate the authors acknowledgment of potential effects of testing behavior in captivity, and I think they justify well the reason that captive individuals are appropriate here.

Thank you.

Line 156 – "number of trials"

This has now been corrected. Thank you for noticing the error.

Figure 1: Exploration could also be measured as the number of visits to incorrect compartments after demonstrating that it has learned the food holding compartment by visiting it at least twice consecutively? i.e. continuing to sample the environment. Although it is unclear whether this occurred.

We did not want the birds to gain experience in the testing chamber while the goal location was no longer rewarded. We therefore released the birds from the testing chamber

immediately after they received the reward. We have edited L173 to try to make this clearer.

Lines 200-201 – Exploration could be measured as the latency from release to first leave the release pen?

This is a good suggestion but our lack of data at the start of the season means that we would not have full confidence in the exact date each individual left the release pen. We therefore do not feel that we should incorporate this.

Lines 205-207 – Can you dumb this down a little bit? Do you mean that you can triangulate the location of the bird from the time it takes the signal to reach each of the base stations during each 5-min increment?

We have tried to make this clearer (L205-211): “Briefly, this system comprised four time-synchronised, fixed-location receiver stations that recorded the time of arrival of individually identifiable radio signals from tags. The tags on our pheasants emitted a signal once every four seconds. The time of the arrival of the signal at each of the receiver stations was used to compute localisations. We then filtered the raw location data (Appendix S2) and computed median locations for each 5 minute period.”

Lines 220-222 – Where do these data come from? Visual observations? The ATLAS tags? This could also be a measure of individual exploration propensity.

We have tried to make this clearer and entirely rephrased this section L216-232 to explain how we determined that the lack of data in the beginning does not affect our conclusions.

Lines 227-228 – Given how fast the 12 spatial learning trials took in captivity I do think this a significant weakness of the study because spatial learning might have occurred and reached an asymptote already by 37 days post release.

The spatial learning trials took 5 days and covered an area 75cm x 75cm, much smaller than the area that our atlas system covers (~3km x 4km). We have tried to provide evidence that the birds are newer to the area *outside* the release pen as mentioned in our responses above and furthermore found that birds increase the straightness of their transitory paths during September, indicating that they are still changing their movements so have not reached an asymptote.

Line 251 – I think it would be good to include the model comparison table in the main text. For example, how are you deciding “best”? Change in AIC? That is important for the reader to interpret the results.

We have tried to make this clearer in the MS. These are not models with completely different covariates but with slightly different initial starting parameters. The differences in starting parameters are very small and we do not think that it would be very informative/interesting to present the initial parameters in the MS itself but we have now included a table describing the initial parameters in the supplementary (S6). We have added a line in the MS to inform the reader that 15/25 of the models resulted in very similar log likelihood ratios which indicates that the initial parameters were good choices as they converged on similar results (L252-253). We have also added a description of how we calculated the best model (largest maximum log-likelihood) on L251.

Line 253-254 – This sentence lacks clarity. The first part of the sentence implies that classifications are inferred from inspecting the movement patterns. So, I’m not sure I’m understanding what the second part of the sentence is trying to convey. Unless I am misinterpreting your intended message, I suggest rewording to simply say “We inspected individual movement patterns to categorize each state.”

Thank you for this feedback, we have edited L259 to try to make this clearer. In brief, once the HMM had objectively categorised the states, we plotted of pheasant paths with each state in a different colour to help us identify which state was which (see Fig 2 for an example). Then we assigned state 1-3 as resting, transit and foraging respectively.

Lines 263-264 – I think the main text of the ms could benefit from the addition of a figure here to illustrate a typical movement path representative of each state. The relevance of turning angles centered around π is difficult to visualize.

Thank you for this suggestion, we agree. We have now added a new figure (Fig 2) to the main text to demonstrate a typical, HMM-classified track.

Line 277 – Are profitable foraging patches the feeders in the release pen? I think it would make sense to tie transit paths directly to the end goal of a feeder or drinker, so that it is clear the movement is goal directed and thus benefitted by efficiency.

With reference to the new figure 2, we hope that it is now clear that we are measuring behaviour (mostly) outside the release pen. We also note that the foraging does not always take place at the feeders themselves. In the example in Fig 2 there is quite a broad foraging bout around the edges of the release pen. We often see birds there foraging in the leaf litter and therefore, in answer to the comment below, we do not think that it is possible to pre-define their navigational goals. Using this objective HMM approach removes the need to assume that birds can only forage at the feeders. We have tried to make this clearer (L296-299)

Line 291 – Even 6 hours in duration seems like a long time to be in one continuous transit phase. Since the mean is 1.7hr, what is your justification for this cutoff?

6 hours is just over the 95th percentile of the data and therefore we have (as objectively as possible) removed the most extreme outliers of our data. We did not want to remove data subjectively because we know that pheasants can be in transit for long periods of time. With our 5-minute resolution of data, it is possible that short (<5 min) foraging bouts may intersperse transitory phases and therefore the birds are still directed in their movements, which we want to capture. The mean is much lower and closer to a value we would expect so we do not feel it is necessary to cut out more data than we already have. However, if the reviewers/editors feel it might be beneficial to modify our cut-off points we would welcome a suggestion of a percentile that they feel would be realistic.

Line 297 – This citation is not in number form, FYI.

Thanks for picking this up. It has now been changed.

Lines 311-312 – Why did the authors dichotomize here? I think more explanation is needed. Otherwise, the comparison of the observed mean number of consecutively correct trials to the distribution of the simulated mean number of consecutively correct trials seems a more intuitive way to get at this question (e.g. was the observed mean higher than expected by chance, indicating birds as a group were not performing randomly).

We originally dichotomised because many of the higher scores are so unlikely to happen by chance that it violates the assumption of the Chi-Square (that no expected value <1). We have since discovered that we can use a Monte Carlo simulation within base R `chisq.test()` to calculate the p-value and test the differences between observed and expected distributions if some bins are <1 . We therefore now present a comparison between the whole distribution of observed and expected scores, rather than dichotomising.

Line 317-318 – was the Gamma error structure a “good” fit, or just the best of the poor fit options?

We found the Gamma distribution to be a “good” fit and it is commonly used to describe movement e.g. step length distributions (e.g. <https://doi.org/10.1002/ece3.4823>) or speed (e.g. <https://doi.org/10.7717/peerj.3745>).

Line 321-322 – Could the authors add a description of why this variable is important to include?

Thank you for the suggestion, we have now added an explanation to L329-331.

Lines 371-373 – I don’t really find this a convincing argument that the birds were still developing spatial memory. Maybe if there was more information provided about the location of the individuals’ transit paths. For example, are any of the transit paths from

outside of the release pen? If so, then I am more convinced that change in straightness and speed of movement reflect some spatial learning. If transit paths are all within the release pen (which is where the base stations are, yes?) then transit paths at the point analyzed could just represent individual differences in activity level.

The base stations (which we have now renamed receiver stations) are not within the release pen but 1-4 km away. They are positioned to give us the best chances of catching movement in the release pen area and the surrounding area. We hope that figure 2 now helps readers to understand that we are not restricted to movement within the release pen but in the surrounding area (where there had been little exploration) and we have edited L206-210 to make this clearer. The point of the sentence mentioned in this comment was to note that the birds had hardly explored the environment before mid-September. We hope that this is now clear.

Line 390 – But peers can be informed individuals if they discover the feeding stations first. Do you have any evidence for or against social information use among the released birds? Is it possible that there was no difference in transit speed of poor performers because they were following a knowledgeable peer that quickly explored the environment and discovered the feeders before the ATLAS tags came online?

We have added to L403-407 to explain why we think this might not be the case. We believe that differences in speed between poor and good performance birds suggests that birds were probably not following each other based on their cognitive scores because then we would expect them to be at similar speeds. If birds follow each other with similar cognitive scores then the results and inferences would not change.

Lines 416-417 – Is it possible for the authors to incorporate satellite imagery to assess factors affecting path selection with step selection functions (i.e.

<https://link.springer.com/article/10.1007/s10980-019-00777-z>

)? This is a direct way to address this piece of the discussion.

While this is an interesting avenue of research, we have tried to answer a specific question: whether performance on a spatial cognition task influences the development of movement,

represented by changes in the speed and straightness of transitory paths. Using hidden Markov models allowed us to separate the state that we felt would be most relevant to this question and therefore only ask about transitory movements. Step selection functions, while a viable method, would not necessarily separate the foraging movements from the transitory movements. There are positives to step selection methods, but we do not think that it is beneficial to completely change our analysis when our current method is also viable.

Line 419 – Are the authors able to use the time + location data to determine if certain individuals move as a group? This could inform an alternative explanation for the results of speed and straightness of movement. There is significant literature indicating that social context influences movement, and that individual differences influence the way in which social context changes movement (<https://pubmed.ncbi.nlm.nih.gov/24047530/> as one example off the top of my head).

While we agree that this might be an important factor, we feel that this is beyond the scope of this manuscript. Adding collective movement into models is not a trivial task and we have instead tried to elaborate on group behaviour in the discussion (L405-409; L434).

Line 450 – There was no effect of cognitive performance on path straightness so the claim that better performers have “more tortuous movement” is not supported by the data

Thank you for picking this up, it was a remnant of a previous version and has now been deleted and the last sentences changed.

Appendix C

RSOS-201758.R1

The authors made few of the revisions I suggested, but they did improve the manuscript by clarifying why they believe the absence of movement data in the initial weeks of release is not problematic for their question. The included figure also significantly adds to the reader's understanding of the data and study region.

However, without incorporating the additional analyses I suggested, namely the consistency with which some individuals use the same movement path or ruling out exploration as an alternative explanation, the conclusion of the paper in terms of movement paths becoming more efficient over time because of spatial memory seems unwarranted. Spatial memory may be responsible for increased path speed if the individuals are only moving within the same areas, but the authors did not add information on this point. How often do individuals use the same routes? How much of the area covered by the ATLAS receivers are individuals consistently using each day? To determine that spatial memory (and not alternatives like exploration/boldness) is the cause of any differential change in movement paths between high and low performers on the spatial cognitive task in captivity, it is necessary to demonstrate that learning is actually able to occur in the natural environment. Simply calculating individual home ranges or the frequency of relocations in small grid cells overlaid across the study area over time could get at this question of whether the pheasants are becoming more familiar with a specific environment and learning efficient paths.

Line edits (based on the track-changes version in the pdf file):

69-82 – Perhaps I was unclear in my first comment, but I still don't see how this paragraph is relevant to the data and hypothesis being tested. The authors are not measuring movement strategies under different contexts, or able to determine the accuracy of spatial decisions. If change in movement behavior over time reflects the development of spatial memory than I am unclear how this relates to a speed/accuracy tradeoff (because speed is increasing as accuracy increases based on learning), so I don't see how this relates. The one mention of speed/accuracy tradeoff in the discussion as an explanation for the results is appropriately brief and speculative and I don't think it justifies a whole paragraph in the introduction.

I reiterate that this paragraph needs to more directly incorporate what is known about the impact of spatial cognition on movement for it to flow with the rest of the paper. In the authors' response to my previous comment, they indicate the paragraph is "a description of the ways animals can collect information and how that can be related to the speed of movement". But this does not come through at all.

Line 99 – this needs to be reworded, perhaps it was supposed to be "efficiency of *an* individual's movement paths..."

Line 198 – I suspect the authors just mean "dusk" here rather than "dawn and dusk".

Line 220 – there is a typo here, "in the how"...

Line 229 – The S4 & S5 figures should probably be included in the main text. That will allow readers to more easily form their own conclusions on whether significant spatial learning was occurring before movement paths were consistently recorded.

Line 264 – what is “sf”?

Appendix D

RSOS-201758.R1

The authors made few of the revisions I suggested, but they did improve the manuscript by clarifying why they believe the absence of movement data in the initial weeks of release is not problematic for their question. The included figure also significantly adds to the reader's understanding of the data and study region.

However, without incorporating the additional analyses I suggested, namely the consistency with which some individuals use the same movement path or ruling out exploration as an alternative explanation, the conclusion of the paper in terms of movement paths becoming more efficient over time because of spatial memory seems unwarranted. Spatial memory may be responsible for increased path speed if the individuals are only moving within the same areas, but the authors did not add information on this point. How often do individuals use the same routes? How much of the area covered by the ATLAS receivers are individuals consistently using each day? To determine that spatial memory (and not alternatives like exploration/boldness) is the cause of any differential change in movement paths between high and low performers on the spatial cognitive task in captivity, it is necessary to demonstrate that learning is actually able to occur in the natural environment. Simply calculating individual home ranges or the frequency of relocations in small grid cells overlaid across the study area over time could get at this question of whether the pheasants are becoming more familiar with a specific environment and learning efficient paths.

RESPONSE:

Thank you for reiterating this point. We hope that we now understand the reviewers concern and in order to address this post hoc, we have calculated individual home ranges for the first half of September and the second half of September. We now show that the overlap of the core home ranges (50%KDE) is high (54%) throughout the month indicating that the birds are able to become familiar with this environment. While the core range overlap is high, the birds have larger home ranges (95%KDE) at the end of September than at the beginning, demonstrating that we are still able to capture the development of movement paths.

We are glad that the reviewer encouraged us to show individual home ranges and this may be helpful to readers that are less familiar with the behaviour of released pheasants, who may have expected that the birds would quickly disperse to completely different areas. We have tried to make this as clear as we can by adding a new figure showing the home ranges of 4 birds at the start (1st-15th sept) and the end (16th -30th) and added text between L213-219: "The mean home range size (95%-Kernel density estimate (KDE) of the pheasants increased over September between 1st -15th September (mean \pm s.d. = 3.97 Ha \pm 3.12) and 16th – 30st September (mean \pm s.d. = 10.69 Ha \pm 12.48), yet the core range (50%- KDE) at the end of the month of still overlapped heavily (mean \pm s.d. = 0.54 \pm 0.18) with the core range from the beginning of the month. This demonstrates that while the birds are beginning to explore the area, they were not dispersing to different completely different areas or moving in a nomadic way. Thus, birds are able to use their previous experience of an area to inform their movements."

We have added the home range figures for all birds for the reviewer only (4 examples are now in the main text as Figure 2), we do not intend to add all of these to the supplementary material but we

can do so if the editor feels that it is necessary. We note that there are a few birds where data was not available at the end of the month. However, the criteria for inclusion in the analysis was to have at least 6 hours of movement data over 7 days. Since the analysis uses time as a continuous variable (not a categorical one as must occur to show home ranges for a particular time period), we do not believe that this is a problem.

Therefore, we now show (by reporting HR sizes over time) that there is the opportunity for learning, according to the reviewer's criteria, namely that individuals are not moving randomly around the landscape but instead are gradually expanding a home range while revisiting the same core areas of that space over time. We cannot rule out the role of boldness/exploration because we did not assess these. Instead, we make inferences on the importance of spatial cognition based on the relationship that we report between an individual's performance in a spatial task (that excluded e.g. boldness/exploration as contributory factors) conducted under controlled conditions early in life, and their differences transitory movement paths in the wild later in life.

Line edits (based on the track-changes version in the pdf file):

69-82 – Perhaps I was unclear in my first comment, but I still don't see how this paragraph is relevant to the data and hypothesis being tested. The authors are not measuring movement strategies under different contexts, or able to determine the accuracy of spatial decisions. If change in movement behavior over time reflects the development of spatial memory than I am unclear how this relates to a speed/accuracy tradeoff (because speed is increasing as accuracy increases based on learning), so I don't see how this relates. The one mention of speed/accuracy tradeoff in the discussion as an explanation for the results is appropriately brief and speculative and I don't think it justifies a whole paragraph in the introduction. I reiterate that this paragraph needs to more directly incorporate what is known about the impact of spatial cognition on movement for it to flow with the rest of the paper. In the authors' response to my previous comment, they indicate the paragraph is "a description of the ways animals can collect information and how that can be related to the speed of movement". But this does not come through at all.

RESPONSE:

We thank the reviewer for reiterating their point. We accept that this paragraph didn't fit quite as well as we had thought. We have now removed it and instead have focussed on how spatial cognition may directly influence the measures that we use in this study. We have edited the introduction heavily, both to address this specific point and, more generally to make the introduction flow better.

Line 99 – this needs to be reworded, perhaps it was supposed to be "efficiency of an individual's movement paths..."

RESPONSE:

Yes, thank you. We have added "an" to this sentence.

Line 198 – I suspect the authors just mean "dusk" here rather than "dawn and dusk".

RESPONSE:

No – we did mean to write dawn and dusk. To clarify for other readers, we now explain that the pheasants can also accidentally fly/glide down from their overnight roost sites at dawn onto the wrong side of the fence. We often found them pacing the fence line to try to get back in, so at both dawn and dusk we guided them back to the entrances as is common practice for game keepers. We have changed the text from. “Until 30th August, we actively guided birds back into the release pen at dawn and dusk if they had flown over the fence while descending from overnight roost sites while moving around the pen during the day.”

Line 220 – there is a typo here, “in the how”...

RESPONSE:

Thank you for noticing this, we have corrected this typo.

Line 229 – The S4 & S5 figures should probably be included in the main text. That will allow readers to more easily form their own conclusions on whether significant spatial learning was occurring before movement paths were consistently recorded.

RESPONSE:

If the editor agrees then we are happy to add these to the main text. For now, we have left them in the supplementary as the new Figure 2 also helps to demonstrate that the birds are exploring outside of the release pen area. We think that this is enough and do not want to overwhelm the text by having 3 figures to explain one section of the methods, but are happy to accept Editorial advice on this.

Line 264 – what is “sf”?

RESPONSE:

We have changed this to “within 0.03”, originally we meant within 0.1 but this has now been changed to a more exact number to demonstrate that there was a very small difference between log likelihoods.